# Carbapenem-Resistant Gram-Negative Bacilli Characterization in a Tertiary Care Center from El Bajio, Mexico

**DOI:** 10.3390/antibiotics12081295

**Published:** 2023-08-08

**Authors:** Jose Raul Nieto-Saucedo, Luis Esaú López-Jacome, Rafael Franco-Cendejas, Claudia Adriana Colín-Castro, Melissa Hernández-Duran, Luis Raúl Rivera-Garay, Karina Senyase Zamarripa-Martinez, Juan Luis Mosqueda-Gómez

**Affiliations:** 1Fellow of the General Directorate of Quality and Education in Health, Ministry of Health, Mexico City 06696, Mexico; 2Department of Medicine and Nutrition, Universidad de Guanajuato, Leon 37670, Mexico; 3Infectious Diseases Laboratory, Infectious Diseases Division, Instituto Nacional de Rehabilitación Luis Guillermo Ibarra Ibarra, Mexico City 14389, Mexico; 4Biology Department, Chemistry Faculty, Universidad Nacional Autónoma de México, Mexico City 04510, Mexico; 5Biomedical Research Subdirection, Instituto Nacional de Rehabilitación Luis Guillermo Ibarra Ibarra, Mexico City 14389, Mexico; 6Hospital Regional de Alta Especialidad del Bajío, Leon 37660, Mexico

**Keywords:** carbapenems, carbapenemase, gram-negative bacteria, drug resistance

## Abstract

Carbapenem-resistant Gram-negative bacilli (CR-GNB) are a major public health concern. We aimed to evaluate the prevalence of CR-GNB and the frequency of carbapenemase-encoding genes in a tertiary referral center from El Bajio, Mexico. A cross-sectional study was conducted between January and October 2022; Gram-negative bacilli (GNB) were screened for in vitro resistance to at least one carbapenem. CR-GNB were further analyzed for carbapenemase-production through phenotypical methods and by real-time PCR for the following genes: *bla*_KPC_, *bla*_GES_, *bla*_NDM_, *bla*_VIM_, *bla*_IMP_, and *bla*_OXA-48_. In total, 37 out of 508 GNB were carbapenem-resistant (7.3%, 95% CI 5.2–9.9). Non-fermenters had higher rates of carbapenem resistance than *Enterobacterales* (32.5% vs. 2.6%; OR 18.3, 95% CI 8.5–39, *p* < 0.0001), and *Enterobacter cloacae* showed higher carbapenem resistance than other *Enterobacterales* (27% vs. 1.4%; OR 25.9, 95% CI 6.9–95, *p* < 0.0001). Only 15 (40.5%) CR-GNB had a carbapenemase-encoding gene; *Enterobacterales* were more likely to have a carbapenemase-encoding gene than non-fermenters (63.6% vs. 30.8%, *p* = 0.08); *bla*_NDM-1_ and *bla*_NDM-5_ were the main genes found in *Enterobacterales*; and *bla*_IMP-75_ was the most common for *Pseudomonas aeruginosa*. The *mcr-2* gene was harbored in one polymyxin-resistant *E. cloacae*. In our setting, NDM was the most common carbapenemase; however, less than half of the CR-GNB showed a carbapenemase-encoding gene.

## 1. Introduction

Antimicrobial resistance (AMR) rates have become a major concern in global health; if the trend continues this way, AMR could become the leading cause of death by 2050 [1]. Carbapenem-resistant *Enterobacterales* (CRE), carbapenem-resistant *Pseudomonas aeruginosa* (CRPA), and carbapenem-resistant *Acinetobacter baumannii* (CRAB) represent high-priority pathogens [2]. The main problem lies in their resistance mechanisms, as they delay appropriate antibiotic therapy, increase hospital length of stay, and worsen patients’ survival [3]. 

The phenomenon of multidrug-resistant (MDR) bacteria has been specially attributed to the indiscriminate use of antibiotics and the presence of antimicrobial resistance genes. Carbapenemases are described as the most frequent mechanism by which carbapenem resistance has increased [4,5]; nevertheless, it is not the only one. Efflux pumps and porin mutations are other mechanisms described in non-carbapenemase-producing CRE, CRPA and CRAB [5,6].

Carbapenemases can belong to three molecular classes according to Ambler: class A, e.g., *Klebsiella pneumoniae* carbapenemase (KPC) enzymes or GES-type enzymes; class B metallo-β-lactamases (MBLs) such as New Delhi MBLs (NDM), Verona integron-encoded MBLs (VIM), or imipenemase (IMP); and class D carbapenem-hydrolyzing oxacillinase (OXA), e.g., OXA-48-like enzymes [7]. 

The rate of carbapenem resistance is considerably higher for non-fermenters (often >60%) than fermenters (often <10%) due to intrinsic factors, including reduced outer membrane permeability and efficient efflux pumps, even in wild-type strains [8]; however, prevalence of carbapenemase-encoding genes varies widely among studies, ranging from 4 to 32% for CRPA, with predominance of *bla*_VIM_, *bla*_IMP_, and *bla*_GES,_ between 45 and 90% for CRE, with clear dominance of *bla*_KPC_ and *bla*_NDM_, and up to 80 to 100% for CRAB, the main genes found being *bla*_OXA-like_ [9]. Global distribution of carbapenemases has shown a predominance of *bla*_KPC_ in North America and Europe; *bla*_NDM_ and *bla*_IMP_ in some regions of Asia; *bla*_VIM_ in Europe; and *bla*_OXA-48_ in the Middle East and Africa [5]. In Mexico, epidemiological data have been expanding in recent years, showing *bla*_NDM_ as the most frequent gene in both *K. pneumoniae* and *E. coli*, followed by *bla*_KPC_ and *bla*_VIM_; among *P. aeruginosa, bla*_VIM_ and *bla*_IMP_ are the main encoded genes, whereas *A. baumannii* has a predominance of *bla*_OXA-24_*, bla*_OXA-40_*,* and *bla*_OXA-23_ [10,11].

As information on carbapenem resistance varies by geographical situation and is constantly evolving, regional surveillance is fundamental for the implementation of adequate infection control measures and provide good antibiotic use policies. We aimed to estimate the prevalence of carbapenem-resistant Gram-negative bacilli (CR-GNB) and characterize the presence of carbapenemase-encoding genes in a tertiary regional referral center from El Bajío, Mexico.

## 2. Results

### 2.1. Gram-Negative Bacilli Isolates 

A total of 508 cultures with positive growth of Gram-negative bacilli (GNB) isolates (including Enterobacterales and non-fermenters) were included; 360 isolates were from inpatient culturing (70.9%). The isolates were recovered from different clinical specimens: urine in 50.8%, blood 15.9%, respiratory tract 15.9%, wound 14.2%, peritoneal fluid 2%, pleural effusion 1%, and 0.2% from cerebrospinal fluid. The strains isolated corresponded to *Escherichia coli* in 53.5%, *Klebsiella pneumoniae* 16.7%, *Pseudomonas aeruginosa* 14.6%, *Enterobacter cloacae* 3.5%, *Proteus mirabilis* 2.4%, *Serratia marcescens* 1.8%, *Morganella morganii* 1.6%, *Acinetobacter baumannii* 1.2%, and *Citrobacter freundii* 1%, whereas the remaining 3.7% was represented by four *Klebsiella oxytoca*, four *Achromobacter xylosoxidans*, three *Providencia rettgeri*, two *Proteus vulgaris*, two *Pantoea agglomerans*; *Raoultella planticola*, *Ochrobactrum anthropi, Klebsiella aerogenes*, and *Citrobacter koseri* contributed with one isolate each. The presence of extended spectrum ß-lactamases (ESBLs) was estimated to be 60.1% (CI 95%, 54.8–65.2); it was identified in 59.9% of *E. coli* and 60.6% of *Klebsiella* spp. 

### 2.2. Carbapenem-Resistant Gram-Negative Bacilli

In total, 37 of 508 strains were identified as carbapenem-resistant isolates, and their frequency was estimated to be 7.3% (95% Confidence Interval [CI] 5.2–9.9); 11/428 were CRE (2.6%, 95% CI 1.3–4.5), and 26/80 were carbapenem-resistant non-fermenters (32.5%, 95% CI 22.4–43.9). Twenty-seven were from inpatient samples (72.9%). Carbapenem-resistant isolates were mainly recovered from urine 29.7%, lower respiratory tract (tracheal or bronchial aspirate) 21.6%, wound 21.6%, blood 16.2%, sputum 5.4%, and peritoneal fluid 5.4%. Carbapenem-resistant isolates corresponded to *P. aeruginosa* in 67.6%, *E. cloacae* 13.5%, *E. coli* 10.8%, *K. pneumoniae* 5.4%, and *A. baumannii* 2.7% (Table 1). 

#### Carbapenem Resistance by Culture Site

According to culture site, carbapenem resistance was higher in tracheal/bronchioalveolar samples (14.5% vs. 6.8%; OR 2.5 95% CI 1.07–5.75, *p* = 0.047) and lower in urine (4.4% vs. 11.6%; OR 0.38 95% CI 0.18–0.79, *p* = 0.009), compared to all other sites. Non-fermenters had higher rates of carbapenem resistance than *Enterobacterales* (32.5% vs. 2.6%; OR 18.3, 95% CI 8.5–39, *p* < 0.0001). Regarding bacterial species, *E. cloacae* showed higher carbapenem resistance than other *Enterobacterales* (27% vs. 1.4%; OR 25.9, 95% CI 6.9–95.8, *p* < 0.0001); among non-fermenters, there was no statistical difference between *P. aeruginosa* and *A. baumannii* (33.8% vs. 16.7%, *p* = 0.657). *E. coli* showed a higher carbapenem resistance rate in blood cultures than other sites (7% vs. 0.4%, OR 17.1 95% CI 1.7–168, *p* = 0.01). All other bacteria did not show significant differences by site of culture. Frequency of CR-GNB is shown in Table 2.

### 2.3. Carbapenemase Production 

First, carbapenemase-producing strains were phenotypically identified by the modified carbapenem inactivation method (mCIM) in 15 of the 37 CR-GNB samples (40.5%); of the latter, twelve were inhibited by the ethylenediaminetetraacetic acid (EDTA) carbapenem inactivation method (eCIM), suggesting Ambler class B carbapenemases, and the remaining three samples were negative to eCIM. Secondly, the NG-test CARBA-5^®^ identified the involved protein in 12 of the 37 CR-GNB samples (32.4%). Lately, carbapenemase-encoding genes were identified by PCR; fifteen isolates (40.5%) were positive for one carbapenemase-encoding gene; and, out of them, seven strains (46.7%) had the bla_NDM_ gene, four (26.7%) the bla_IMP_ gene, three (20%) the bla_GES_ gene, and one (6.6%) the bla_VIM_ gene. No bla_OXA-like_ or bla_KPC_ genes were detected. There were no co-producer genes. *Enterobacterales* showed a higher rate of carbapenemase-encoding genes than non-fermenters (63.6% vs. 30.8%); however, this result did not reach statistical significance (*p* = 0.08).

All results were concordant between NG-test CARBA-5^®^ and PCR, except for bla_GES_, which is not included in the NG-test CARBA-5^®^. After the DNA sequencing, out of the bla_NDM_ genes, bla_NDM-1_ and bla_NDM-5_ were the most prevalent, with 42.9% each, followed by bla_NDM-7_ in 14.2%. Among bla_IMP_ genes, there were three bla_IMP-75_ (75%) and one bla_IMP-83_ (25%); furthermore, there were two bla_GES-40_, one bla_GES-26,_ and, finally, one bla_VIM-2_. The only isolate of *E. cloacae* resistant to colistin was positive for the *mcr-2* gene (Table 3).

### 2.4. Antimicrobial Resistance Rates

Resistance to ertapenem was 100%, meropenem 89%, doripenem 89%, and imipenem 84%. Only one *E. cloacae* showed resistance to colistin. Among these strains, the resistance to at least one aminoglycoside was 57%, 62% for monobactams, 68% for fourth-generation cephalosporins, 49% for fluoroquinolones, and 62% for piperacillin–tazobactam (Table 4).

## 3. Discussion

Globally, many regions have not implemented active surveillance programs to monitor CR-GNB. Our work is the first to characterize carbapenemase genes in the Bajio region of Mexico, expanding the knowledge of the mechanisms of resistance in our country. We found an overall carbapenem-resistance rate in Gram-negative bacteria of 7.3% (95% CI, 5.2–9.9), being 2.6% (95% CI 1.4–3.5) for *Enterobacterales* and 32.5% (95% CI 22.4–43.9) for non-fermenters, which is comparable with data from Mexico but lower than other international reports [4,9,12]. 

The University Plan for the Control of Antimicrobial Resistance (PUCRA, *Plan Universitario de Control de la Resistencia Antimicrobiana*), a national surveillance program, showed a carbapenem resistance rate of 0.4–1% for *E. coli*, 1–5% for *K. pneumoniae*, 2–4% for *Enterobacter* sp., 59% for *Acinetobacter* sp., and 25% for *P. aeruginosa* [13]. On a subsequent study by the INVIFAR network in Mexico [11], they reported a low carbapenem resistance in *E. coli* (<1%) and *K. pneumoniae* (2.9%) but high carbapenem resistance in *E. cloacae* (10.9%), *P. aeruginosa* (38–44%), and *A. baumannii* (83%). Our center showed similar rates for *E. coli* (1.5%), *K. pneumoniae* (2.4%), and *P. aeruginosa* (33.8%) but a higher prevalence of carbapenem resistance for *E. cloacae* (33%); although it was lower for *A. baumannii* (16.7%), it should be noted that the latter was underrepresented in our study. In addition, the INVIFAR network reported that 45% of *E. coli* and 39% of *K. pneumoniae* were ESBLs producers, which were much lower compared with our rates of approximately 60%. 

The Antimicrobial Surveillance Program SENTRY in Latin America showed that 4.3% of *Enterobacterales* strains were CRE, with the highest rates in Brazil (9%) and Argentina (6.3%) and the lowest in Chile (0.4%) and Mexico (0.7%) [14]. However, these numbers have evolved in recent years, and our study showed a CRE rate of 2.6%. The GLASS report in 2021 mentioned a worldwide prevalence of CRE up to 10–15% [15]. However, it should be noted that Mexico was not part of the antimicrobial resistance surveillance programs included in this study. On the other hand, the prevalence of carbapenem resistance among non-fermenters in Latin America has been described as high as 66% for *P. aeruginosa* and up to 90% for *A. baumannii* [16], discordant to the findings of our study.

Carbapenem resistance is directly influenced by the culture site, being higher in respiratory infections than in bloodstream infections [17]. This was concordant with our analysis, where the highest proportion of CR-GNB were isolated from lower respiratory tract samples, whereas the lowest proportion was for urine cultures. 

Given the high lethality rate associated with CRE, CRPA, and CRAB, assessment of the hospital- or ward-level risk of CR-GNB is crucial, especially by defining the specific resistance mechanisms [3]. However, few laboratories can perform routine phenotypic or genotypic testing for carbapenemases; thus, determining the local epidemiology allows for a better choice of treatment [18]. The latter becomes relevant due to the possible use of the new, although not widely available, β-lactam/β-lactamase inhibitors, such as ceftazidime/avibactam, meropenem/vaborbactam, and imipenem/relebactam, against KPC-producing strains [3,18,19]. Nevertheless, the emergence of resistance to ceftazidime/avibactam has been recently described in KPC-3-producing strains [20]; ceftazidime/avibactam or cefiderocol as monotherapy are options against OXA-producing strains [18,21,22]. For MBL-producing strains, there are more limited options, especially the combination of ceftazidime/avibactam with aztreonam or cefiderocol as monotherapy, whereas tigecycline, eravacycline, and polymyxins remain as alternative but more toxic options, regardless of the presence of carbapenemases [18,23]. This becomes a challenge for Mexico since it is a country with a predominance of MBLs [10,11].

The most frequent types of carbapenemase-encoding genes in our study were *bla*_NDM_ in *Enterobacterales* and *bla*_IMP_ in *P. aeruginosa*; however, no *bla*_OXA_ or *bla*_KPC_ were identified. Worldwide, *bla*_NDM_ and *bla*_KPC_ are the main carbapenemases found in *Enterobacterales*;, whereas *bla*_OXA_, *bla*_IMP_, *bla*_VIM,_ and *bla*_GES_ are primarily found in *P. aeruginosa* and *A. baumannii* [5,24,25,26]. Few studies in Mexico have evaluated the presence of carbapenemase-encoding genes in CRE, CRPA, and CRAB. For CRPA, *bla*_IMP_ and *bla*_VIM_, followed by *bla*_GES_, are the most commonly described [10,11,27,28], in line with our results. Meanwhile, for CRE, some Mexican authors have described *bla*_NDM-1_ as the most associated gene; however, *bla*_OXA-48_, *bla*_OXA-181_, *bla*_OXA-232,_ and *bla*_KPC-2_ have also been reported [10,11,29,30,31]. In addition to *bla*_NDM-1_, we identified isolates carrying the *bla*_NDM-5_ and *bla*_NDM-7_ genes. For CRAB, the most frequent encoding genes described were *bla*_OXA-24_ and *bla*_OXA-40_, followed by *bla*_OXA-23_, *bla*_VIM,_ and *bla*_NDM_ [11]. It is noteworthy that we had a high proportion of isolates with no carbapenemase-encoding genes (60%) in contrast to previously reported studies [4,12,25,32], suggesting the presence of non-enzymatic mechanisms or carbapenemases other than the ones evaluated, closer to those described by Garza-Gonzalez et al. [11], where 47% was negative for the presences of the carbapenemases tested.

The use of point-of-care tests, such as NG-test CARBA-5^®^, is a good complementary option to guide antimicrobial therapy since they have shown excellent performances compared with molecular techniques and provide results within 15–30 min. However, it should be considered that they do not detect all types of carbapenemases, as seen with *bla*_GES_ in our sample, and may not perform as well for *P. aeruginosa* [33]. 

In addition to allowing better guidance of antibiotic therapy, distinguishment between carbapenemase-mediated and non-enzymatic mechanisms is important since the former has been associated with higher odds of dying at 14 days and can spread more efficiently between patients, requiring an efficient equipment disinfection policy, combined with timely isolation of the cohort to reduce the spread of these strains [9,34]. However, as non-carbapenemase-producing strains not only target carbapenems and β-lactam antibiotics, they usually are associated with MDR. Mutations in AmpC and its regulatory genes are commonly attributed to *P. aeruginosa* resistance to new β-lactam-β-lactamase inhibitor combinations [35], whereas plasmid-encoded AmpC is most commonly found in resistant *Enterobacterales* [36]. Among efflux-pump systems, AcrAB-TolC RND is the most commonly described [37]. Finally, porin mutations (especially in OmpC and OmpF) block the entry of carbapenems into the cell [38]. 

Polymyxin susceptibility remains in almost all cases, except for one *E. cloacae* with the presence of *mcr-2* gene. Despite their high toxicities, polymyxins continue as a last-line pharmacological group in cases of Gram-negative MDR infections. The *mcr* genes are of great concern due to their high potential for horizontal propagation. Among the variants described, *mcr-1* is the most prevalent within Mexico [10] and globally [39]; thus far, *mcr-2* has been described in human specimens, especially in Asia [40,41]. However, to the best of our knowledge, this is the first report of *mcr-2* in a Mexican patient.

## 4. Materials and Methods

### 4.1. Design

A cross-sectional study was carried out between 1 January and 31 October 2022 at the Hospital Regional de Alta Especialidad del Bajío, a 184-bed tertiary regional referral center located in León, Guanajuato, Mexico.

### 4.2. Sampling

Isolates from clinically relevant sites samples, such as blood, urine, respiratory, cerebrospinal fluid, peritoneal, pleural, and deep-tissue cultures with Gram-negative bacterial growth, were included. All data were collected in the microbiology laboratory at the Hospital Regional de Alta Especialidad del Bajío. Identification was performed with VITEK-2 (Biomérieux, Marcy l’ Etoile, France), and susceptibility tests were performed by microdilution broth method following the Clinical and Laboratory Standards Institute (CLSI) 2018 M07 recommendations [42] and breakpoints according to M100 Ed. 32 (2022) criteria [43]. Duplicated isolates, i.e., more than one isolate per patient in the same hospitalization, were identified and discharged of the database. Bacteria with resistance to at least one carbapenem (meropenem, imipenem, doripenem, and ertapenem; this last one was not considered for *Pseudomonas aeruginosa*) were selected for analysis. Bacteria from single colonies were stored at −20 °C in brain heart infusion broth with 15% of glycerol until used. An internal protocol (HRAEB-POE-013) for type B biological substances (UN 3373) was used to send the samples to the infectious disease laboratory at the *Instituto Nacional de Rehabilitación Luis Guillermo Ibarra Ibarra* in Mexico City. 

### 4.3. Tests

We performed two phenotypic tests to screen for carbapenemase-producing isolates. First, we performed mCIM (modified carbapenemase inactivation method). To identify metallo ß-lactamase-producing strains, we used the mCIM variant, with EDTA (ethylenediaminetetraacetic acid) as a chelator (eCIM) [42]. Once the producer strain was identified, the next step was to identify the protein involved through NG-test CARBA-5^®^ [33] (KPC, OXA-48, VIM, IMP, and NDM). (eCIM). All these isolates were further analyzed by a polymerase chain reaction (PCR) to identify the following carbapenemase-encoding genes: *bla*_KPC_, *bla*_GES_, *bla*_NDM_, *bla*_VIM_, *bla*_IMP_, and *bla*_OXA-48_. An *Enterobacter cloacae* was shown to be resistant to colistin; therefore, it was decided to be screened for mobile colistin resistance (*mcr*) genes. The gene sequences used are listed in Table 5.

#### 4.3.1. Carbapenemase Related-Genes Detection

Bacterial DNA was extracted by thermal shock (95 °C/20 min) using PBS 1×. Next, the tubes were centrifuged (10,000 rpm/20 min), and the supernatants were separated in new tubes. PCR conditions were the following: 10× PCR Buffer (New England BioLabs, Ipswich, MA, USA) containing 2 mM of MgCl_2_, 0.2 mM of each of the dNTPs (Invitrogen, Waltham, MA, USA), 10 pmol of each oligonucleotide (Table 5), 1.5 U of Taq polymerase (BioLabs, USA), and 3 uL of DNA.

The amplification was carried out using the following program: 1 cycle of 5 min at 95 °C, 35 cycles of 50 s at 95 °C, 60 s at 55 °C, and 50 s at 68 °C, followed by 1 final cycle of 10 min at 68 °C (Veriti, Applied Biosystems, Waltham, MA, USA); Tm for *bla*_IMP_ was 52 °C, Tm *bla*_oxa-48_, *bla*_VIM_, *bla*_GES_, *bla*_NDM_, and *bla*_KPC_ was 55 °C; Tm for *mcr-1* and *mcr-2* was 54 °C.

PCR products were separated in a 1% agarose gel at 120 volts for 40 min, and SYBRGreen (Invitrogen, USA) was used such as DNA intercalant. Bands were visualized in Gel Doc XR+ (BioRad, Hercules, CA, USA) with Image Lab Software version 6.1 (BioRad, Hercules, CA, USA).

#### 4.3.2. Sequencing

Prior to the sequencing process, re-amplification and labeling using the BigDye^®^ Direct Cycle Sequencing Kit (Thermo Scientific, Waltham, MA, USA) were performed. Sequencing was performed using the two oligonucleotides (described for the amplification). A 3730xl DNA Analyzer (Applied Biosystems, USA) was used for sequencing, using POP-7 (Thermo Scientific, USA). GenBank was used to compare the sequences obtained with those of reference organisms in this database [44]. Identification was corroborated by performing independent sequencing for each primer.

### 4.4. Statistical Analysis

SPSS software version 25.0 was used for statistical analysis. A descriptive analysis of the distribution of all Gram-negative strains at the hospital was performed, as well as an analysis of the subpopulation of carbapenem-resistant Gram-negative strains. Groups were established according to the site where the sample was obtained and the microorganism isolated. The absolute and relative frequencies for each group and their relationships with the antibiotic resistance profile and the presence of carbapenemases were described. A comparative analysis between groups was performed. *p*-values were obtained using the chi-square test (*x*^2^) or Fisher’s exact test for qualitative variables, according to their expected values. For quantitative variables, the Student’s *t* test or Mann–Whitney U test was used according to their distribution, which was evaluated by the Shapiro–Wilk test. The confidence interval will be 95%, with a *p*-value required for statistical significance <0.05. 

### 4.5. Limitations

Despite being a regional referral center, our results may not be applicable in primary care settings, as our patients were more likely to have previous hospitalizations and antibiotic exposures due to their underlying conditions. We did not evaluate all the described carbapenemase-encoding genes. Also, porin and efflux-pump mutations were not assessed; therefore, we could not attribute with certainty the mechanism of resistance in some strains. As we did not evaluate our population throughout time, future research should focus in this direction to analyze the CR-GNB trends in our country.

## 5. Conclusions

In the region of El Bajio, Mexico, the prevalences of carbapenem-resistant strains are 2.6% for *Enterobacterales* and 32.5% for non-fermenters. Less than half of CR-GNB showed a carbapenemase-encoding gene. *bla*_NDM_ was the most frequent carbapenemase-encoding gene for *Enterobacterales*, whereas *bla*_IMP-75_ was for *P. aeruginosa*. Continuous surveillance is necessary to provide adequate control measures and improve antibiotic stewardship. *E. cloacae* in our setting will be of special attention since we reported the first Mexican isolate harboring the *mcr-2* gene.

## Figures and Tables

**Table 1 antibiotics-12-01295-t001:** Origin and bacterial differences between carbapenem-resistant and susceptible isolates.

	Carbapenem-Susceptible Isolates (*n* = 471)	Carbapenem-Resistant Isolates (*n* = 37)	*p*-Value
Age, years (IQR)	48 (27–61)	43 (27–55)	0.405 *
Sex, *n* (%)			0.004 **
Male	214 (45.4)	26 (70.3)	
Female	257 (54.6)	11 (29.7)	
Setting, *n* (%)			0.770 **
Inpatient	333 (70.7)	27 (73)	
Outpatient	138 (29.3)	10 (27)	
Culture site, *n* (%)			
Urine	247 (52.4)	11 (29.7)	0.007 **
Blood	75 (15.9)	6 (16.2)	1 **
Sputum	24 (5.1)	2 (5.4)	1 ***
Tracheal/bronchial	47 (10)	8 (21.6)	0.047 ***
Wound	64 (13.6)	8 (21.6)	0.177 **
Peritoneal fluid	8 (1.7)	2 (5.4)	0.160 ***
Pleural fluid	5 (1.1)	0	1 ***
CSF	1 (0.2)	0	1 ***
*Enterobacterales*, *n* = 428 (%)	417 (88.5)	11 (29.7)	
*E. coli*, *n* = 272	268 (56.9)	4 (10.8)	0.107
*K. pneumoniae*, *n* = 85	83 (17.6)	2 (5.4)	1
*E. cloacae*, *n* = 18	13 (2.8)	5 (13.5)	<0.0001
Others, *n* = 53	53 (11.2)	0	0.373
Non-fermenters, *n* = 80 (%)	54 (11.5)	26 (70.3)	0.657
*P. aeruginosa*, *n* = 74	49 (10.4)	25 (67.6)	
*A. baumannii*, *n* = 6	5 (1)	1 (2.7)	

Comparison between carbapenem-susceptible and carbapenem-resistant isolates. CSF = Cerebrospinal fluid; IQR = Interquartile range. * Mann–Whitney U test. ** Chi-square test. *** Fisher-exact test.

**Table 2 antibiotics-12-01295-t002:** Frequency of carbapenemase resistance in Gram-negative according to culture site.

Bacteria	Culture Site						Overall Estimation of Carbapenem-Resistance Prevalence
	Urine	Blood	Sputum	Tracheal/Bronchial	Wound	Peritoneal Fluid	
*Enterobacterales*							
*E. coli*	0/182	3/43	0/6	0/7	1/28	0/3	4/272
	(0%)	(7%)	(0%)	(0%)	(3.6%)	(0%)	(1.5%)
*K. pneumoniae*	1/30	0/13	0/6	0/18	0/12	1/4	2/85
	(3.3%)	(0%)	(0%)	(0%)	(0%)	(25%)	(2.4%)
*E. cloacae*	1/3	1/2	0/1	1/6	2/6	0	5/18
	(33.3%)	(50%)	(0%)	(16.7%)	(33.3%)		(27.8%)
Others	0/23	0/9	0/2	0/6	0/11	0/2	0/53
Estimated prevalence for *Enterobacterales*	2/238	4/67	0/15	1/37	3/57	1/9	11/428
	(0.8%)	(6.0%)	(0%)	(2.7%)	(5.2%)	(11.1%)	(2.6%)
Non-fermenters							
*P. aeruginosa*	9/19	2/12	2/11	6/15	5/15	1/1	25/74
	(47.4%)	(16.7%)	(18.2%)	(40%)	(33.3%)	(100%)	(33.8%)
*A. baumannii*	0/1	0/2	0	1/3	0	0	1/6
	(0%)	(0%)		(33.3%)			(16.7%)
Estimated prevalence for non-fermenters	9/20	2/14	2/11	7/18	5/15	1/1	26/80
	(45%)	(14.2%)	(18.2%)	(38.9%)	(33.3%)	(100%)	(32.5%)
All							
Overall estimation prevalence according to site	11/258	6/81	2/26	8/55	8/72	2/10	37/508
	(4.3%)	(7.4%)	(7.7%)	(14.5%)	(11.1%)	(20%)	(7.3%)

Prevalence of carbapenem-resistant bacilli by culture site and species.

**Table 3 antibiotics-12-01295-t003:** Carbapenemase-encoding genes by bacterial species.

Bacterial Specie	Isolates (%)	Carbapenemase Genes				Carbapenemase Producing Isolates (%)
		Class A	Class B			
		GES	NDM	IMP	VIM	
*Enterobacterales*						
*E. cloacae*	5 (13.5)	-	2 NDM-1	-	-	2 (40)
*E. coli*	4 (9.5)	-	3 NDM-5 1 NDM-1	-	-	4 (100)
*K. pneumoniae*	2 (5.4)	-	1 NDM-7	-	-	1 (50)
Total of CRE	11 (29.7)	-	7 (100)	-	-	7 (63.6)
Non-fermenters						
*P. aeruginosa*	25 (67.6)	2 GES-40 1 GES-26	-	3 IMP-75 1 IMP-83	1 VIM-2	8 (32)
*A. baumannii*	1 (2.7)	-	-	-	-	0
Total of CRPA and CRAB	26 (70.3)	3 (37.5)	-	4 (50)	1 (12.5)	8 (30.8)
All						
Total (%)	37 (100)	3 (8.1)	7 (18.9)	4 (11)	1 (2.7)	

Genotype detection of carbapenemase-encoding genes in Gram-negative bacilli. CRE = carbapenem-resistant *Enterobacterales*, CRPA = carbapenem-resistant *Pseudomonas aeruginosa*, CRAB = carbapenem-resistant *Acinetobacter baumannii.*

**Table 4 antibiotics-12-01295-t004:** Antimicrobial resistance of the carbapenem-resistant isolates.

*P. aeruginosa* (*n* = 25)													
	AMK I = 32 R ≥ 64	GEN I = 8 R ≥ 16	ATM I = 16 R ≥ 32	CAZ I = 16 R ≥ 32		CEF I = 16 R ≥ 32	CIP I = 1 R ≥ 2	LVX I = 2 R ≥ 4	DOR I = 4 R ≥ 8	IMP I = 4 R ≥ 8	MEM I = 4 R ≥ 8	COL I ≤ 2 R ≥ 4	PTZ I = 32/4–64/4 R ≥ 128/4
I	16	12	8	4		8	8	4	0	0	4	100	36
R	24	60	52	52		44	44	40	96	88	92	0	36
***E. cloacae* (*n* = 5)**													
	AMK I = 32 R ≥ 64	GEN I = 8 R ≥ 16	ATM I = 8 R ≥ 16	CAZ I = 8 R ≥ 16	CEF SDD = 4–8 R ≥ 16	CIP I = 0.5 R ≥ 1	LVX I = 1 R ≥ 2	ETP I = 1 R ≥ 2	DOR I = 2 R ≥ 4	IMP I = 2 R ≥ 4	MEM I = 2 R ≥ 4	COL I ≤ 2 R ≥ 4	PTZ SDD = 16/4 R ≥ 128/4
I	0	0	0	0	0	20	0	0	0	0	0	80	0
R	20	40	80	100	40	20	20	100	60	60	60	20	100
***E. coli*** **(*n* = 4)**													
	AMK I = 32 R ≥ 64	GEN I = 8 R ≥ 16	ATM I = 8 R ≥ 16	CAZ I = 8 R ≥ 16	CEF SDD = 4–8 R ≥ 16	CIP I = 0.5 R ≥ 1	LVX I = 1 R ≥ 2	ETP I = 1 R ≥ 2	DOR I = 2 R ≥ 4	IMP I = 2 R ≥ 4	MEM I = 2 R ≥ 4	COL I ≤ 2 R ≥ 4	PTZ SDD = 16/4 R ≥ 128/4
I	0	0	25	0	0	0	0	0	0	0	0	100	0
R	25	25	75	100	75	75	75	100	100	100	100	0	100
***K. pneumoniae*** **(*n* = 2)**													
	AMK I = 32 R ≥ 64	GEN I = 8 R ≥ 16	ATM I = 8 R ≥ 16	CAZ I = 8 R ≥ 16	CEF SDD = 4–8 R ≥ 16	CIP I = 0.5 R ≥ 1	LVX I = 1 R ≥ 2	ETP I = 1 R ≥ 2	DOR I = 2 R ≥ 4	IMP I = 2 R ≥ 4	MEM I = 2 R ≥ 4	COL I ≤ 2 R ≥ 4	PTZ SDD = 16/4 R ≥ 128/4
I	0	0	0	0	0	0	50	0	0	0	0	100	0
R	50	50	50	100	50	100	50	100	50	50	50	0	100
***A. baumannii*** **(*n* = 1) ***													
	AMK I = 32 R ≥ 64	GEN I = 8 R ≥ 16	CAZ I = 16 R ≥ 32	CEF I = 16 R ≥ 32		CIP I = 2 R ≥ 4	LVX I = 4 R ≥ 8	DOR I = 4 R ≥ 8		IMP I = 4 R ≥ 8	MEM I = 4 R ≥ 8	COL I ≤ 2 R ≥ 4	PTZ I = 32/4–64/4 R ≥ 128/4
I	0	0	0	0		0	0	0		0	0	100	0
R	0	100	100	100		100	100	100		100	100	0	100

Results are expressed in resistance percentage (%) according to CLSI 2022. All minimal inhibitory concentration breakpoints are expressed in μg/mL. I: Intermediate; R: Resistant; SDD: susceptible-dose-dependent. Antibiotics tested: AMK: amikacin; ATM: aztreonam; CAZ: ceftazidime; CEF: cefepime; CIP: ciprofloxacin; COL: colistin; DOR: doripenem; ETP: ertapenem; GEN: gentamicin; IMP: imipenem; LVX: levofloxacin; MEM: meropenem; PTZ: piperacillin/tazobactam. * Only 1 isolate of *A. baumannii* was included.

**Table 5 antibiotics-12-01295-t005:** Gene sequences used for polymerase chain reaction.

Gen	Sequence	Amplicon (bp)
*bla* _GES_	Forward 5′-TCATTCACGCHCTATTVCTGGCA-3′ Reverse 5′-CTATTTGTCCGTGCTCAGG-3′	857
*bla* _KPC_	Forward 5′-ATGTCACTGTATCGCCGTCT-3′ Reverse 5′-TTACTGCCCGTTGACGC-3′	798
*bla* _NDM_	Forward 5′-ATGGAATTGCCCAATATT-3′ Reverse 5′-TCAGYGCAGCTTGTCGGC-3′	650
*bla* _VIM_	Forward 5′-AGATTGVCGATGGTGTTTGGT-3′ Reverse 5′-GAGCAAGTCTAGACCGCCC-3′	430
*bla* _IMP_	Forward 5′-GTTTATGTTCATACTTCGTTTG-3′ Reverse 5′-CAACCAGTTTTGCHTTAC-3′	425
*bla* _OXA-48_	Forward 5′-GAATGCCTGCGGTAGCAA-3′ Reverse 5′-AAACCATCCGATGTGGGCAT-3′	438
*mcr-1*	Forward 5′-TCTTGTGGCGAGTGTTGCCGT-3’ Reverse 5′-CCAATGATACGCATGATAAACGCTG-3′	190
*mcr-2*	Forward 5′-CTGTTGCTTGTGCCGATTGGACTA-3′ Reverse 5′-ACGGCCATAGCCATTGAACTGC-3′	210

## Data Availability

All data are available upon request to the corresponding author.

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
