# Peer review of "Carbapenem-Resistant Gram-Negative Bacilli Characterization in a Tertiary Care Center from El Bajio, Mexico"

_antibiotics, 2023, doi:10.3390/antibiotics12081295_

Round 1

Reviewer 1 Report

Very generic introduction on the issue of multidrug-resistant bacteria, it does not focus at all on the subject of the article, it should be more concrete about it. 

Lines 56-60, explain that non-fermenting bacilli are much more resistant to carbapenems per se intrinsically, even in their wild-type strains. 

Lines 61-66, add something about the most frequent carbapenemases in the study area, including bibliography. 

Results:

Remember to always put the bacterial species in italics. 

Line 89, do not use "swab", it is more correct to use wound or tissue exudate.

Table 1: The p-value column is difficult to interpret, it is necessary either to make a new column or to give more details. 

From line 110: explain why carbapenemases have been detected with different methods, develop which ones have been applied in a reasoned way. Again, take care to correctly italicise bacterial species. 

Table 4: simplify it visually, it appears too much "μg/mL" and makes it unattractive to consult. 

Discussion: 

Line 240, does not specify how antibiotic sensitivity was tested. 

Furthermore, I think the title is not the most appropriate, as it does not fully characterise the carbapenemases studied. They are detected by PCR, named and nothing more. 

Author Response

  1. The introduction has been modified, with emphasis on global and national epidemiology.
  2. Lines 57 and 58: It has been explained that nonfermenters are intrinsically more resistant than Enterobacterales even in wild-type strains.
  3. Lines 64 to 68: The most frequent carbapenemases in our country were added, as well as their references.
  4. All species have been italicized.
  5. "Swab" has been changed to "wound" in all cases.
  6. Table 1 has been reorganized to be as clear as possible with their P values. 
  7. Line 119: The sequential order of the tests and their detailed explanation has been described in the methods section.
  8. Table 4 has been simplified in the units of measurement and antibiotics tested.
  9. The method used for antibiotic sensitivity was the broth microdilution test, the specification was added in the methods.
  10. A slight modification has been proposed to the title, changing "molecular characterization" to only "characterization". 

Reviewer 2 Report

Line 67 and beyond - Italics are missing in all species

Line 81- "(CI 95%, 54.8-65.2);" what means CI? 

Line 92- Something is not quite right in Table 1. Example: P. aeruinosa resistant 25 (67.6) and susceptible 49 (10.4). First nowhere in the table does it indicate that the values in parentheses are percentages. I was the one who had to deduce from the text above. Second, R + S=78%. And the remaining 22%? Intermediates? It must be somewhere. I, who work with EUCAST, only work with R or S, not intermediaries. must make some note.....

line 108- Table 2- The tables are not at all intuitive.

line 123 to Table 3- Confusing with the results... According to the materials and methods, for the genes, nowhere is it written or the primers on which variants of each gene are going to be studied. So the technique used was going to be PCR and then they already talk about Sequencing? what type was it, nanopore, lumina??

Line 150. "Gram negative " - Gram-negative 

Line 240 - different font. 

Linne 241 - CLSI: You need to add the year- Every year the standards change. 

Line 244- "Bacteria with resistance at least one carbapenem (meropenem, imipenem, doripenem and ertapenem, t his last one was not considered for Pseudomonas aeruginosa). " - why

Line 260 and all text. "mcr1"- correcrt for mcr-1 and mcr-2. Respect the nomenclature.

Line 270 to 272 - different font.

Line 283- Conclusions too poor. Please modify .

Line 288 - Author contributions - too confusing, just put the initials for easy interpretation, For example: Jose Raul Nieto-Saucedo replace with J.R.N-S.

More general considerations:

The materials and methods do not agree with some of the results obtained, not only in the example I gave above but in more.

Why did you decide to study the MCR gene if the purpose of the article was " Carbapenem-resistant Gram-negative bacilli"?

The discussion is too vague. What do you think these results bring back? What is your study for? Did they detect Carbapenem-resistant Gram-negative bacilli and then what to do?

Author Response

  1. All species have been italicized.
  2. The meaning of the abbreviation "CI" has been specified as a confidence interval.
  3. Table 1: In the first column, the unit of measurement is specified in parentheses. 
    The number of strains in each column (2 and 3) was divided according to their susceptibility or resistance to carbapenems (intermediates were included in resistant strains, as specified in methods). The number of strains was added for each individual specie, e.g. Pseudomonas aeruginosa had 74 isolates, 49 susceptible and 25 resistant. However, their percentages are expressed with respect to the total number of susceptible and resistant strains, 49/471 (10.4) and 25/37 (67.6), therefore the sum is not 100% in each line.
  4. It was not possible to simplify Table 2, since its objective was to describe the resistance rate for species and culture site. 
  5. Table 3 and methods, section 4.3: A detailed description of the PCR and sequencing techniques used has been added.
  6. Line 161: "Gram-negative", corrected.
  7. Only one font has been used.
  8. Lines 256-257: The CLSI edition and year have been specified.
  9. Lines 259-261: The fact of not including ertapenem for Pseudomonas aeruginosa was due to its intrinsic resistance, in accordance with the CLSI standards, so we preferred not to add it to the text.
  10. Nomenclature about mcr-1 and mcr-2 has been corrected.
  11. The conclusions were modified, starting with the prevalence of carbapenem resistance in our region, followed by the frequency of carbapenemases and their type. What this type of study is useful for us and, finally, the finding of the strain carrying the mcr-2 gene is emphasized.
  12. Author contributions have been abbreviated.
  13. The methods used have been described in more detail to be consistent with the results presented.
  14. Although it was not the main object of the study, the reason for investigating the presence of the mcr gene was that during the analysis a strain resistant to colistin was found, being of relevance for local and national epidemiology.
  15. The discussion has been modified.
    Our study is the first of its kind in the Mexican Bajío region, expanding epidemiological knowledge. It allows improving antibiotic optimization programs in our region. The usefulness of point-of-care testing to improve the initiation of appropriate therapy is highlighted, as well as the therapeutic options available according to the type of carbapenemase and the infection control measures that should be implemented due to its rapid dissemination and higher mortality.

Reviewer 3 Report

The paper by Nieto-Saucedo et al. describes the characterization of carbapenem-resistant Gram-negative bacilli in Mexico.  Overall, the paper does a good job outlining the characterization, but there are a few things that need to be addressed.

1.  In the results section Lines 69-82, none of the species are italicized.

2.  Line 84 Thirty-seven of what?

3.  Table 1 P values do not line up properly and what is the value in parenthesis?

4.  Table 4 What does the number identified for I and R represent?

5.  For the PCR, no reference is cited, no controls are mentioned, no PCR conditions are noted, and the number of replicates is not mentioned.

Just minor English language corrections need to be made.

Author Response

  1. All species have been italicized.
  2. Line 91: Thirty-seven "of 508 strains".
  3. P values have been simplified according to their individual values, and values comparing two groups have been left only in the main text.
  4. The numbers in Table 4 for I (intermediate) and R (resistant) represent percentages, this is specified in the table footnote.
  5. A more detailed description of the PCR methods has been added. Sequences were obtained from GenBank, reference has been added.

Round 2

Reviewer 1 Report

For my part, I think that the errors that I saw have been correctly modified.

Reviewer 2 Report

Accept in present form

Reviewer 3 Report

The authors have addressed my concerns.

Minor edits